# Prevalence of metabolic syndrome and associated factors among patients with chronic Chagas disease

Isis Gabrielli Gomes Xavier[1], Marcelo Carvalho Vieira[1,2], Luiz Fernando Rodrigues Junior[3], Gilberto Marcelo Sperandio da Silva[1], Paula Simplicio da Silva [1], Marcelo Teixeira de Holanda[1], Erica Rodrigues Maciel[1], Fernanda Martins Carneiro[1], Flavia Mazzoli-Rocha[1], Luiz Henrique Conde Sangenis[1], Fernanda de Souza Nogueira Sardinha Mendes[1], Alejandro Marcel Hasslocher-Moreno [1], Andrea Silvestre de Sousa[1], Andrea Rodrigues da Costa[1], Roberto Magalhães Saraiva[1], Pedro Emmanuel Alvarenga Americano do Brasil[1], Mauro Felippe Felix Mediano [1,3]*

1 Evandro Chagas National Institute of Infectious Disease, Oswaldo Cruz Foundation, Rio de Janeiro, RJ, Brazil, 2 Center for Cardiology and Exercise, Aloysio de Castro State Institute of Cardiology, Rio de Janeiro, RJ, Brazil, 3 Department of Research and Education, National Institute of Cardiology, Rio de Janeiro, RJ, Brazil

* mauro.mediano@ini.fiocruz.br

**Data Availability Statement:** All data files are available from the Open Science Framework database (https://osf.io/9ev8b/).

## Abstract

The increase in life expectancy and the migration of individuals with Chagas disease (ChD) from rural to urban centers exposes them to the development of chronic-degenerative abnormalities that may increase the prevalence of metabolic syndrome (MetS). The present study aimed to identify the prevalence of MetS and its components in individuals with chronic ChD. This is a cross-sectional study with 361 patients of both sexes, aging >18 years, followed at a national reference center (Rio de Janeiro, Brazil). MetS diagnosis followed the International Diabetes Federation 2005 criteria. The association between the variables was determined through logistic regression models. The mean age was and 60.7 ±10.8 years. About half (56.2%) were female and the majority self-reported their race as mulatto (59.8%). The percentage of individuals with MetS was 40.4%. The variables independently associated with MetS were age (OR 1.06; 95%CI 1.04–1.09), high education levels (OR 0.36; 95%CI 0.17–0.79) and cardiac form with heart failure (OR 0.34; 95%CI 0.17–0.68). Therefore, a high prevalence of MetS was found in this Brazilian chronic ChD cohort. The identification of the associated factors can facilitate the development of effective approaches for preventing and managing MetS in ChD patients.

## Introduction

Chagas disease (ChD) is a neglected tropical disease caused by the protozoan *Trypanosoma cruzi* with 8 million people estimated to be infected worldwide [1–3]. Initially restricted to Latin America, it is currently widespread over several countries in almost all continents, being considered a global epidemic [1, 4]. The increase in life expectancy together with migration of

**Funding:** This study was funded by Fundação de Amparo a Pesquisa Carlos Chagas (grant number 111.133/2014) and by the Coordination for the Improvement of Higher Education Personnel (Coordenação de Aperfeiçoamento de Pessoal de Nível Superior - CAPES) - Finance Code 001.

**Competing interests:** The authors have declared that no competing interests exist.

large part of the ChD population from rural to urban areas increased the exposure of these patients to inadequate lifestyle, facilitating the development of non-infectious chronic conditions such as obesity, insulin resistance, hypertension and dyslipidemia [5]. Together, these factors exponentially increase the risk of cardiovascular events and death, leading to the development of an important clinical condition known as metabolic syndrome (MetS) [6].

MetS was initially described by Reaven as a cluster of clinical and metabolic abnormalities including glucose intolerance, hypertension, dyslipidemia, and insulin resistance that have in the visceral fat accumulation as a common pathway [7, 8]. Currently, MetS is defined by the World Health Organization (WHO) as a pathological condition characterized by central obesity, insulin resistance, hypertension and hyperlipidemia [9].

The prevalence of MetS has been growing at an alarming rate over the past decades, reaching up to 40% of the entire population, depending on the diagnostic criteria and the studied population [10]. This high prevalence is particularly concerning since individuals with MetS have two to threefold risk of developing cardiovascular disease and a fivefold risk of developing diabetes [11]. However, the percentage of patients with ChD presenting MetS is still unknown and the deleterious effects of MetS together with the clinical ChD-related abnormalities may further decrease quality of life and increase health-related costs, morbidity and mortality rates [12, 13]. Thus, studies aiming to identify the factors associated to MetS in ChD patients are of paramount importance and could facilitate the development of effective approaches for preventing and managing MetS in ChD patients.

Therefore, the present study aimed to investigate the prevalence of MetS in individuals with chronic ChD as well as to identify the main associated factors related to this clinical syndrome in this population.

## Methods

### Study design, period and population

This is an observational cross-sectional study, conducted from March 2014 to March 2017, including residents in the city of Rio de Janeiro of both sexes, aging >18 years, with diagnosis of ChD confirmed by two simultaneously positive serological tests (enzyme-linked immunosorbent assay and indirect immunofluorescence). All patients were under follow-up at the outpatient center of the national reference center for treatment and research in infectious and tropical diseases in Rio de Janeiro, Brazil (Evandro Chagas National Institute of Infectious Disease/ Oswaldo Cruz Foundation). Participants were excluded if they presented autoimmune disorders, cancer, other infectious diseases at the time of study recruitment, non-Chagasic heart diseases, severe cognitive impairments that precluded the completion of the questionnaires, current use of chronic anti-inflammatory or corticosteroids, or pregnant.

### Sample size

The sample size was calculated based on a previous study conducted in a Brazilian urban center that achieved a 30% prevalence of MetS [14]. Considering a precision of 5% and 95% confidence interval, 323 individuals were necessary to perform this study. The sample size was further increased by 20% to account for refusals, totalizing a sample of 400 individuals.

### Ethical considerations

All participants received information about the goals and procedures of the study and agreed to participate by signing an informed consent form. The study was approved by the

Institutional Review Board of the Evandro Chagas National Institute of Infectious Disease (CAAE: 58273916.0.000.5262).

## Study procedures

Patients were invited to participate during their regular clinic visits and were submitted to the study procedures in two visits within a period of no more than two months. In the first visit, patients signed the informed consent, completed all the questionnaires, and performed anthropometric and blood pressure measurements whereas in the second visit they underwent a clinical evaluation and blood tests. Trained staffs administered the questionnaires and performed the anthropometric and blood pressure measurements. The same physician performed the clinical evaluation in all participants. Blood samples were draw in the morning after a 12-hour fasting.

## Metabolic syndrome

MetS was defined following the criteria established by the International Diabetes Federation in 2005 as the presence of central obesity, measured as ethnic-specific increased waist circumference (for South American population $\geq$ 90 cm in men and $\geq$ 80 cm in women), plus at least two of the following components: 1) raised triglycerides ($\geq$150 mg/dL or specific treatment for this lipid abnormality); 2) reduced HDL-cholesterol (<40 mg/dL in males and <50 mg/dL in females or specific treatment for this lipid abnormality); 3) raised resting blood pressure (systolic blood pressure $\geq$130 mmHg or diastolic blood pressure $\geq$85mmHg or treatment of previously diagnosed hypertension); 4) raised fasting plasma glucose (glucose$\geq$100mg/dL or previously diagnosed type 2 diabetes mellitus) [15].

## Clinical form of ChD

Patients were classified using clinical, electrocardiographic, echocardiographic, and digestive exams according to the presence of ChD related abnormalities into indeterminate, cardiac without heart failure, cardiac with heart failure or digestive forms following the Brazilian Consensus on Chagas Disease [16].

## Comorbidities

Comorbidities (hypertension, diabetes, dyslipidemia, and obesity) were obtained using information from medical records and anthropometric measures during the clinical evaluation. Obesity was diagnosed if body mass index [BMI = weight (kg)/squared height (m$^2$)] was $\geq$ 30 kg/m$^2$. Blood pressure measurements were taken twice with participants seated in a quiet room after 5 minutes rest using an Omron® digital sphygmomanometer and the mean value was considered.

## Socioeconomic data and lifestyle

Information on age, sex, schooling, race, residents by domicile and income *per capita* were obtained during the interviews. Age was calculated subtracting date of interview from date of birth and considered as a continuous variable. Schooling was categorized based on the formal years of study into <9 years, 9 to 12 years and >12 years. Race was self-reported and classified as white, black, mulatto and others. The income *per capita* was obtained summing up all income from each resident in the domicile and dividing by the number of residents [17]. Smoking, alcohol consumption, sleep duration, physical activity level and food intake were evaluated during the interviews. Smoking was classified as current (regular use of tobacco,

regardless of how long), former (past occasional use of tobacco for at least 3 months or daily use for a period of at least 1 month) or non-smoker (currently does not use any tobacco product that emits smoke, even occasionally, even if have experienced) [18]. Alcohol consumption was categorized into none (never ingested alcohol during life), former (did not consume any amount of alcohol in the last 30 days, having ingested in the past) or current (consumed any amount of alcohol in the last 30 days). Sleep hours was determined through a direct question and treated as a continuous variable. Physical activity levels were determined using the validated Brazilian short version of the International Physical Activity Questionnaire (IPAQ-short) [19, 20]. This instrument comprises eight questions regarding the duration and frequency of vigorous, moderate, and light physical activity, allowing individuals to be classified into three different categories: mild, moderate and high. Food consumption was assessed using a 24-hour recall that consists on the identification and quantification of all food and beverages consumed in the day before the interview [21]. Macronutrients were calculated using DietWin Professional Version 2008 software.

### Data management and statistical analysis

Exploratory data analysis was performed calculating means (standard deviations) and frequency (percentages) of the variables of interest. The association between MetS and exposure variables was determined using logistic regression models. A univariate logistic regression was performed to determine the variables that should be included in the multivariate model, that included only those with $p < 0.20$ in the univariate analysis. The backwards method was used to sequentially remove variables with p-values greater than 0.05 in the multivariate analysis, until the final model that maintained only those with $p < 0.05$. The Research Electronic Data Capture (REDCap) web application was used for data management and the data analysis was conducted using Stata 13.0 software. Statistical significance was set at $p \leq 0.05$ for all analyses.

## Results

From 397 included patients, 36 were excluded due to the following reasons: 6 with other infectious diseases, 3 with auto-immune diseases, 6 with cancer, 8 with non-chagasic cardiomyopathy, 5 in use of anti-inflammatory or corticosteroids, and 8 did not return to the second visit (losses to follow-up). Therefore, the final sample consisted of 361 individuals. The overall mean age was 60.7 (±10.8) years, with 56.2% (n = 203) women. There was a predominance of mulatto race (59.8%; n = 216) and most participants had less than 9 years of schooling (67.3%; n = 243). The percentage of individuals diagnosed with MetS was 40.4% (n = 146). The prevalence of hypertension, dyslipidemia, obesity and diabetes were 67.3% (n = 243), 53.5% (n = 193), 25.8% (n = 93) and 21.6% (n = 78), respectively (Fig 1).

The description of the variables stratified by the presence of MetS is shown in Table 1. Overall, MetS patients were older (64.6 vs 58.1; p<0.001), with a predominance of females (63.7% vs 51.2%; p = 0.02) and less than 9 years of education (74.7% vs 62.3%; p = 0.004). The minority of MetS patients presented cardiac form with heart failure (8.2% vs 20.5%; p = 0.002) but had a greater prevalence of comorbidities (2.7 vs 0.99; p<0.001), and higher levels of triglycerides (127.4 vs 105.5; p<0.001), very low-density lipoprotein (25.4 vs 20.5; p<0.001), glucose (108.1 vs 97.4; p<0.001), and systolic blood pressure (139.2 vs 129.1; p<0.001). The consumption of carbohydrates (181.1 vs 204.7; p = 0.009) and lipids (36.3 vs 41.2; p = 0.020) was lower among MetS patients.

The univariate analysis showed a significant association between MetS and age (OR 1.06; 95% CI 1.04 to 1.09), female sex (OR 1.67; 95% CI 1.09 to 2.58), education levels > 12 years (OR 0.30; 95% CI 0.14 to 0.63), cardiac form with heart failure (OR 0.35; 95% CI 0.18 to 0.69),

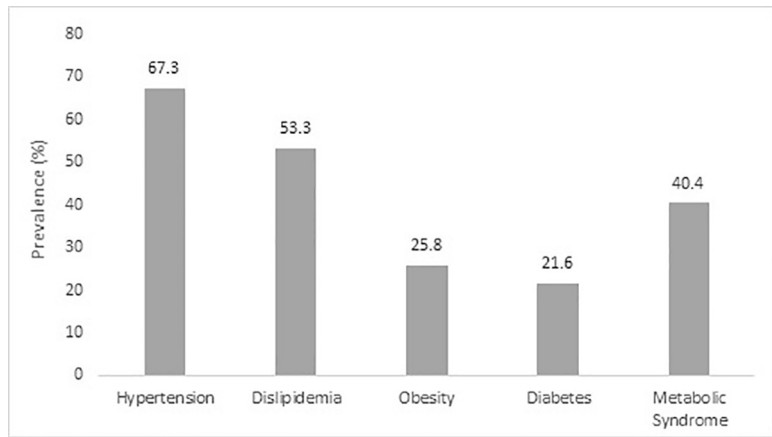

**Fig 1.**

and the consumption of carbohydrate (OR 0.99; 95% CI 0.99 to 0.99) and lipids (OR 0.99; 95% CI 0.98 to 1.00) (Table 2). In the multivariate model, the variables that were independently associated with MetS were age (OR 1.06; 95% CI 1.04 to 1.09), education levels > 12 years (OR 0.36; 95% CI 0.17 to 0.79) and cardiac form with heart failure (OR 0.34; 95% CI 0.17 to 0.68) (Table 3).

## Discussion

The main finding of the present study was a high prevalence (about 40%) of MetS in patients with chronic ChD that was greater than in the general population in other studies conducted in Brazil and worldwide [22, 23]. For instance, a study including 1.663 individuals from a random sample of an overall Brazilian urban adult population found a lower MetS prevalence (about 30%) in comparison to our study [14]. Moreover, a study including 137 Latin American migrants that were diagnosed with ChD at the Geneva University Hospitals found a MetS prevalence of 16.8% [24]. In Europe and United States, the MetS prevalence widely ranged from 20 to 60% depending on the classification criteria and the characteristics of the studied population (e.g, age, sex, and race) [25–27]. In this context, a cross-sectional analysis including 243 older patients (> 60 years) found a high MetS prevalence (more than 60%) using the same criteria than our study (IDF 2005), suggesting that age could be considered an important risk factor for MetS [28]. Similarly, a study including data from the National Health and Nutrition Survey also found an increased prevalence of MetS with aging, that varied from 18% in the 2nd decade of life to 50% after age 60 [27]. Therefore, the elevated prevalence of MetS found in the present study could be attributed to the aging of Chagas disease patients over the last decades, as previously demonstrated by others [12, 29]. In our outpatient cohort, the mean age raised from 45 to 61 years over the last two decades, reinforcing the aging pattern of this population, especially for those that live in urban areas where the disease transmission is quite low and the access and quality of healthcare services related to ChD improved over the last years [13].

The prevalence of comorbidities in our study was high, confirming previous findings in ChD patients of our group that described similar frequencies of hypertension (56%), dyslipidemia (42%), and diabetes (30%) in 619 ChD patients, with most of them (more than 70%) presenting at least one of the aforementioned comorbidities [30]. Others also showed an elevated prevalence of comorbidities in Brazilians (57% of hypertension, 20% of dyslipidemia, 10% of diabetes mellitus) [12] and Bolivians (64% of hypertension, 67% of obese or overweight) [31]

**Table 1. Characteristics of participants included in the study (n = 361).**

| Variables | Metabolic Syndrome | | p-value[*] |
|---|---|---|---|
| | **No** | **Yes** | |
| | **(60.6%; n = 215)** | **(40.4%; n = 146)** | |
| Age (years) | 58.1 (±11.7) | 64.6 (±7.9) | <0.001 |
| Residents by domicile (persons) | 2.8 (±1.35) | 2.8 (±1.60) | 0.91 |
| Income *per capita* (per R$1000.00) | 905.7 (±1013.5) | 954.9 (±7745) | 0.62 |
| Sex (%) | | | |
| Male | 48.8 (105) | 36.3 (53) | 0.02 |
| Female | 51.2 (110) | 63.7 (93) | |
| Race (%) | | | |
| White | 25.1 (54) | 18.5 (27) | 0.21 |
| Black | 12.1 (26) | 16.4 (24) | |
| Mulatto | 60.0 (129) | 59.6 (87) | |
| Others | 2.8 (6) | 5.5 (8) | |
| Schooling (%) | | | |
| < 9 years | 62.3 (134) | 74.7 (109) | 0.004 |
| 9–12 years | 18.6 (40) | 18.5 (27) | |
| >12 years | 19.1 (41) | 6.9 (10) | |
| Sleep duration (hours) | 6.5 (±1.57) | 6.8 (±1.62) | 0.08 |
| SBP (mmHg) | 129.1 (±22.4) | 139.2 (±19.8) | <0.001 |
| DBP (mmHg) | 75.7 (±13.4) | 77.8 (±10.4) | 0.11 |
| Comorbidities (%) | | | |
| Hypertension | 48.4 (104) | 95.2 (139) | <0.001 |
| Diabetes Mellitus | 8.8 (19) | 40.4 (59) | <0.001 |
| Dyslipidemia | 29.8 (64) | 88.4 (129) | <0.001 |
| Obesity | 12.1 (26) | 45.9 (67) | <0.001 |
| Medication | | | |
| Antihypertensive | 69.8 (150) | 94.5 (138) | <0.001 |
| Hypoglycemic | 7.4 (16) | 21.2 (31) | <0.001 |
| Hipolipemic | 23.7 (51) | 61.6 (90) | <0.001 |
| Number of comorbidities (%) | 0.99 (± 0.78) | 2.7 (± 0.72) | <0.001 |
| Biomarkers | | | |
| Total Cholesterol (mg/dL) (n = 355) | 182.9 (±35.5) | 187.2 (±37.4) | 0.267 |
| Triglycerides (mg/dL) (n = 354) | 105.5 (±65.2) | 127.4 (±56.4) | <0.001 |
| HDL-cholesterol (mg/dL) (n = 304) | 51.7 (±14.5) | 50.0 (±14.7) | 0.538 |
| LDL-cholesterol (mg/dL) (n = 303) | 113.3 (±30.0) | 113.1 (±35.9) | 0.969 |
| VLDL-cholesterol (mg/dL) (n = 352) | 20.5 (±11.2) | 25.4 (±11.3) | <0.001 |
| Glucose (mg/dL) (n = 360) | 97.4 (±18.4) | 108.1 (±37.0) | <0.001 |
| Glycated Hemoglobin (%) (n = 296) | 6.1 (±1.0) | 6.3 (±1.0) | 0.052 |
| C-reactive protein (mg/L) (n = 275) | 0.44 (±1.4) | 0.55 (±1.6) | 0.574 |
| Smoking (%) | | | |
| Non-smoker | 53.9 (116) | 52.1 (76) | 0.929 |
| Former | 40.5 (87) | 41.8 (61) | |
| Current | 5.6 (12) | 6.2 (9) | |
| Alcohol consumption (%) | | | |
| None | 61.9 (133) | 58.2 (85) | 0.776 |
| Former | 14.9 (32) | 15.8 (23) | |
| Current | 23.3 (50) | 26.0 (38) | |

(*Continued*)

**Table 1.** (Continued)

| Variables | Metabolic Syndrome | | p-value[*] |
|---|---|---|---|
| | **No** | **Yes** | |
| | **(60.6%; n = 215)** | **(40.4%; n = 146)** | |
| Physical activity level (%) | | | |
| Low | 25.6 (55) | 26.0 (38) | 0.834 |
| Moderate | 47.0 (101) | 49.3 (72) | |
| High | 27.4 (59) | 24.7 (36) | |
| Indeterminate form | 26.6 (56) | 28.1 (41) | 0.67 |
| Cardiac form without heart failure | 50.7 (109) | 58.9 (86) | 0.13 |
| Cardiac form with heart failure | 20.5 (44) | 8.2 (12) | 0.002 |
| Digestive form | 16.3 (35) | 15.8 (23) | 0.89 |
| Caloric consumption (Kcal) | 1279.3 (±722.8) | 1161.4 (±643.8) | 0.101 |
| Macronutrients (g) | | | |
| Carbohydrate | 204.7 (±89.4) | 181.1 (±84.1) | 0.009 |
| Protein | 69.8 (±34.3) | 64.2 (±30.3) | 0.123 |
| Lipid | 41.2 (±22.2) | 36.3 (±18.2) | 0.020 |
| Fibers | 18.9 (±10.5) | 17.6 (±12.1) | 0.214 |

[*] Unpaired t-test for continuous and chi-squared test for categorical variables.

Means (standard deviation) for continuous and percentage (absolute frequency) for categorical variables.

NYHA- New York Heart Association; SBP- systolic blood pressure; DBP- diastolic blood pressure; HDL- high-density lipoprotein; LDL- low-density lipoprotein; VLDL- very low-density lipoprotein.

patients with ChD. The alarming high prevalence of comorbidities among ChD patients could be explained by the migration from rural to urban areas, that improved life expectancy but also increased the exposure to inadequate lifestyles, such as unhealthy eating habits and decreased physical activity levels.

Three variables were independently associated with MetS as follows: age, educational level, and clinical form of ChD. As previously discussed, the prevalence of MetS increases with age, with prevalence of 10% in individuals aged 20 to 29 years, 20% in individuals between 40 and 49 years and 45% in individuals between 60 and 69 years [32]. Nonetheless, the high prevalence of individuals with MetS may indicate the changes in the characteristics of the ChD population over the last decades and that non-ChD comorbidities could potentially impact their health, deserving more attention.

Higher educational levels (>12 years) were associated with 64% lower odds of having MetS in comparison to < 9 years of education. Similarly, a cohort study conducted with 1.915 Korean adults found an inverse association between educational level and MetS, suggesting that socioeconomic disparities may increase the risk of MetS [33]. To our knowledge, no previous study examined the influence of socioeconomic variables on the risk of MetS among patients ChD. People with high educational level tend to be healthier than those with low educational level [34], as they usually present a better socioeconomic status, an important health determinant, and have more general knowledge and practice of healthy lifestyles (dietary habits, physical activity, nonsmoking, mental health), that contribute to a better control of the comorbidities that characterize MetS [35, 36].

In the present study, individuals that presented cardiac form with heart failure had a lower prevalence of MetS. A possible explanation for this finding is that advanced heart failure is associated with significant weight loss, due to the high state of catabolism and cachexia,

**Table 2. Univariate logistic regression for the association between MetS and exposure variables in patients with chronic Chagas disease (n = 361).**

| Variables | | *Odds Ratio* | 95%CI | *p-value* |
|---|---|---|---|---|
| Age (years) | | 1.06 | 1.04–1.09 | <0.001 |
| Sex (female) | | 1.67 | 1.09–2.58 | 0.02 |
| Residents by domicilie (persons) | | 0.99 | 0.86–1.15 | 0.91 |
| Income *per capita (*per R$ 1000.00) | | 1.00 | 1.00–1.00 | 0.62 |
| Race | | | | |
| | White | Reference | Reference | Reference |
| | Black | 1.85 | 0.90–3.80 | 0.09 |
| | Mulatto | 1.35 | 0.79–2.31 | 0.27 |
| | Others | 2.67 | 0.84–8.46 | 0.09 |
| Schooling | | | | |
| | <9 years | Reference | Reference | Reference |
| | 9–12 years | 0.83 | 0.48–1.44 | 0.51 |
| | >12 years | 0.30 | 0.14–0.63 | <0.001 |
| Sleep duration (hours) | | 1.13 | 0.99–1.29 | 0.08 |
| Smoking (%) | | | | |
| | Non-smoker | Reference | Reference | Reference |
| | Former | 1.07 | 0.69–1.66 | 0.76 |
| | Current | 1.14 | 0.46–2.85 | 0.77 |
| Alcohol consumption (%) | | | | |
| | None | Reference | Reference | Reference |
| | Former | 1.12 | 0.62–2.05 | 0.70 |
| | Current | 1.19 | 0.72–1.96 | 0.50 |
| Physical activity level (%) | | | | |
| | Low | Reference | Reference | Reference |
| | Moderate | 1.03 | 0.62–1.72 | 0.90 |
| | High | 0.88 | 0.49–1.59 | 0.68 |
| Indeterminate form | | 1.11 | 0.69–1.78 | 0.67 |
| Cardiac form without heart failure | | 1.39 | 0.91–2.13 | 0.13 |
| Cardiac form with heart failure | | 0.35 | 0.18–0.69 | 0.002 |
| Digestive form | | 0.96 | 0.54–1.71 | 0.89 |
| Carbohydrates (g) | | 0.99 | 0.99–0.99 | 0.01 |
| Protein (g) | | 0.99 | 0.99–1.00 | 0.11 |
| Lipids (g) | | 0.99 | 0.98–1.00 | 0.03 |
| Fibers (g) | | 0.99 | 0.97–1.01 | 0.23 |
| Caloric consumption (kcal) | | 1.00 | 1.00–1.00 | 0.12 |

**Table 3. Multivariate logistic regression for the association between MetS and exposure variables in patients with Chagas disease (n = 361).**

| Variables | | *Odds Ratio* | 95%CI | *p-value* |
|---|---|---|---|---|
| Age (years) | | 1.06 | 1.04–1.09 | <0.001 |
| Schooling | | | | |
| | <9 years | Reference | Reference | Reference |
| | 9–12 years | 0.91 | 0.51–1.62 | 0.75 |
| | >12 years | 0.36 | 0.17–0.79 | 0.01 |
| Cardiac form with heart failure | | 0.34 | 0.17–0.68 | 0.003 |

decreasing body fat deposition [37]. Moreover, individuals with heart failure are encouraged to closely self-manage their illness, monitoring signs and symptoms (e.g, weight gain), and better complying with medical regimens and lifestyle recommendations to optimize health outcomes and quality of life [38, 39].

Surprisingly, carbohydrate consumption was a protective factor for the development of MetS in the univariate analysis (OR 0.99; 95% CI 0.99 to 1.00; p < 0.01), although not reaching statistical significance in the multivariate model (OR 0.99; 95%CI 0.99 to 1.00; p = 0.06). Studies evaluating food consumption in patients with ChD are scarce. In a case-control study including 81 patients with ChD and 81 controls, Castilhos et al. [40] evaluated food and nutrients intake among ChD patients followed in a tertiary hospital. Similar to our results, the prevalence of obesity and increased waist circumference was lower among ChD patients, but with a higher intake of carbohydrates. The lack of information about the quality of the carbohydrate consumed in the present study could explain this unexpected finding in which a greater consumption of high-quality carbohydrates (not only the quantity) is associated with a lower risk of MetS [41]. Moreover, since patients included in the present study are followed in a national reference center and received a comprehensive care treatment including a multidisciplinary approach, it is possible that those patients had been previously identified with metabolic abnormalities during their routine clinic visits and had initiated nutritional assistance before the study procedures, a classical example of reverse causation in cross-sectional studies. In this case, longitudinal studies are necessary to better elucidate the influence of macronutrients consumption on the risk of MetS in patients with ChD.

Another unexpected result was the lack of association between physical activity levels and MetS. Although largely used in epidemiological studies, questionnaire is not the golden standard measure of physical activity which may have increased measurement error, especially when applied in a sample characterized by very low educational levels, leading to nondifferential misclassification [42, 43].

The present study has some limitations. Our sample consisted of patients regularly monitored at a national reference center which may represent a selection bias, limiting the external validity. Moreover, these results should not be extrapolated to general population, since participants with ChD has specific characteristics. Moreover, the cross-sectional design prevents us from making conclusions about the causal relationship between MetS and the variables investigated. On the other hand, this is the first study evaluating the prevalence of MetS and its main associated factors that included a relatively high sample size of patients with chronic ChD. Finally, multiple comparison tests from different linear regression models can increase the probability of type 1 error, even though our results were consistent after decreasing the probability of type 1 error to 1% (all variables had p-values <0.01 in the multivariate model).

To conclude, in the present study we found a high prevalence of MetS in patients with chronic ChD while hypertension was the most prevalent comorbidity. The variables independently associated with MetS were age, education level, and clinical form of ChD (more specifically, heart failure). In this setting, the identification of patients'characteristics associated to MetS can facilitate the development of effective approaches (e.g lifestyle modifications such as nutritional counseling and physical exercise) for preventing and managing this syndrome in ChD patients.

## Acknowledgments

The authors thank the Evandro Chagas National Institute of Infectious Disease for the clinical and logistical support.

## Author Contributions

**Conceptualization:** Isis Gabrielli Gomes Xavier, Pedro Emmanuel Alvarenga Americano do Brasil, Mauro Felippe Felix Mediano.

**Data curation:** Isis Gabrielli Gomes Xavier.

**Formal analysis:** Isis Gabrielli Gomes Xavier, Mauro Felippe Felix Mediano.

**Funding acquisition:** Mauro Felippe Felix Mediano.

**Investigation:** Marcelo Carvalho Vieira, Luiz Fernando Rodrigues Junior, Gilberto Marcelo Sperandio da Silva, Paula Simplicio da Silva, Marcelo Teixeira de Holanda, Erica Rodrigues Maciel, Fernanda Martins Carneiro, Fernanda de Souza Nogueira Sardinha Mendes, Alejandro Marcel Hasslocher-Moreno, Pedro Emmanuel Alvarenga Americano do Brasil, Mauro Felippe Felix Mediano.

**Methodology:** Isis Gabrielli Gomes Xavier, Marcelo Carvalho Vieira, Luiz Fernando Rodrigues Junior, Gilberto Marcelo Sperandio da Silva, Paula Simplicio da Silva, Marcelo Teixeira de Holanda, Erica Rodrigues Maciel, Fernanda Martins Carneiro, Luiz Henrique Conde Sangenis, Alejandro Marcel Hasslocher-Moreno, Andrea Rodrigues da Costa, Pedro Emmanuel Alvarenga Americano do Brasil, Mauro Felippe Felix Mediano.

**Project administration:** Roberto Magalhães Saraiva, Mauro Felippe Felix Mediano.

**Resources:** Roberto Magalhães Saraiva, Mauro Felippe Felix Mediano.

**Supervision:** Isis Gabrielli Gomes Xavier, Luiz Henrique Conde Sangenis, Fernanda de Souza Nogueira Sardinha Mendes, Andrea Silvestre de Sousa, Andrea Rodrigues da Costa, Mauro Felippe Felix Mediano.

**Validation:** Mauro Felippe Felix Mediano.

**Visualization:** Flavia Mazzoli-Rocha, Mauro Felippe Felix Mediano.

**Writing – original draft:** Isis Gabrielli Gomes Xavier, Marcelo Carvalho Vieira, Luiz Fernando Rodrigues Junior, Gilberto Marcelo Sperandio da Silva, Paula Simplicio da Silva, Marcelo Teixeira de Holanda, Erica Rodrigues Maciel, Luiz Henrique Conde Sangenis, Fernanda de Souza Nogueira Sardinha Mendes, Alejandro Marcel Hasslocher-Moreno, Andrea Silvestre de Sousa, Roberto Magalhães Saraiva, Mauro Felippe Felix Mediano.

**Writing – review & editing:** Isis Gabrielli Gomes Xavier, Marcelo Carvalho Vieira, Luiz Fernando Rodrigues Junior, Gilberto Marcelo Sperandio da Silva, Paula Simplicio da Silva, Marcelo Teixeira de Holanda, Erica Rodrigues Maciel, Fernanda Martins Carneiro, Flavia Mazzoli-Rocha, Luiz Henrique Conde Sangenis, Fernanda de Souza Nogueira Sardinha Mendes, Alejandro Marcel Hasslocher-Moreno, Andrea Silvestre de Sousa, Andrea Rodrigues da Costa, Roberto Magalhães Saraiva, Pedro Emmanuel Alvarenga Americano do Brasil, Mauro Felippe Felix Mediano.

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
