## [Decision Letter · Decision Letter 0]

6 Nov 2020

PONE-D-20-16190

Prevalence of metabolic syndrome and associated factors among patients with chronic Chagas disease

PLOS ONE

Dear Dr. Mediano,

Thank you for submitting your manuscript to PLOS ONE. After careful consideration, we feel that it has merit but does not fully meet PLOS ONE’s publication criteria as it currently stands. Therefore, we invite you to submit a revised version of the manuscript that addresses the points raised during the review process.

We look forward to receiving your revised manuscript.

Kind regards,

Venkatachalam Udhayakumar

Academic Editor

PLOS ONE

Journal Requirements:

2. Thank you for stating in your manuscript test that "All participants received information about the goals and procedures of the study and agreed to participate by signing an informed consent form." Please also add this information to your ethics statement in the online submission form.

4. Please include a caption for figure 1.

Additional comments to address:

Abstract: Line 9 starts with “Most participants were women (56.2%)”. The use of the term “Most” can be replaced with a different term as women participants are only slightly higher than men.

Data analysis: Since there were multiple comparisons, wondering if there was an attempt to correct for this by adjusting the P values and significance threshold?

Results and Discussion: Majority of the study participants were from Mulatto ethnic group. It will be useful to discuss weather a high prevalence (40%) of MetS in patients with chronic ChD identified in this population is generalizable to Brazilian population broadly.

In the last part of the discussion it will be useful to state how this study finding can be helpful in developing preventive strategies to minimize the risk of Mets in ChD patients.

Reviewers' comments:

Reviewer's Responses to Questions

**Comments to the Author**

1. Is the manuscript technically sound, and do the data support the conclusions?

Reviewer #1: Yes

2. Has the statistical analysis been performed appropriately and rigorously? 

Reviewer #1: Yes

3. Have the authors made all data underlying the findings in their manuscript fully available?

Reviewer #1: Yes

4. Is the manuscript presented in an intelligible fashion and written in standard English?

Reviewer #1: Yes

5. Review Comments to the Author

Reviewer #1: Prevalence of metabolic syndrome and associated factors among patients with chronic

Chagas disease

1. Overview of the manuscript: The manuscript describes the prevalence of the metabolic syndrome and its components in patients with chronic Chagas disease and the possible increase of the MetS in individuals that migrated from rural to urban centers. The authors reported a high prevalence (about 40%) of MetS in patients with chronic ChD which is greater than the general population reported in other studies that were conducted in Brazil and worldwide. However, it is important to take in consideration the age of the population, degree of education, the study design etc. The Metabolic syndrome is normally associated with increased risk for cardiovascular disease that it is directly related to the lifestyle. The rates can be different based on the country, individual sites, study criteria classification and the characteristics of the studied population. The authors evaluated the metabolic syndrome and its components using clinical electrocardiographic, echocardiographic, and digestive exams. No pathological aspects of the Chagas diseases were mentioned, especially in the patients that developed the MetS.

Overall, this seems to be a well conducted study, the results are generally clearly presented, and the paper is well written. The purpose of this study is relevant. There are not major issues in interpreting the study data. However, in order to improve the quality of the manuscript, the issues outlined below should be considered before publication.

2. Introduction section: For a better comprehension of the study, below are a few suggestions:

a. It will be better if the manuscript had line numbers and page references.

b. In the first paragraph of the introduction, reference 1 refers to an article from 2015 and reference 2 refers to an article from 2018 (Médecins Sans Frontières). I suggest using more updated information, such as the WHO website.

c. The MetS is mentioned for the first time in the second paragraph. I suggest reviewing this paragraph and better describe the syndrome, including a short history of the background and how there has been varying definitions until the WHO provided the currently recognized international definition.

3. Methods section: There are few suggestions which follow:

1. If the patients are from different sites or a specific site in Brazil, it would beneficial to provide a map of the study site(s).

2. I would recommend separating the methods into:

a. Study design, period and population

b. Sample size.

c. Study procedure or Inclusion criteria.

d. Ethical considerations

e. Clinical follow up

f. Clinical form of ChD

g. Evaluation of nutritional status

h. Socioeconomic data and lifestyle

i. Data management and statistical analysis

4. Results section:

a. Please specify in the text where this data can be found (table?) “The overall mean age was 60.7 years, with 56.2% women. There was a predominance of mulatto race (59.8%) and most participants had less than 9 years of schooling (67.3%)”. If it is listed in the table 1, please review the values, they are different from table 1.

b. The percentage regarding “The prevalence of hypertension, dyslipidemia, obesity and diabetes were 67.3% (n=243), 53.5% (n=193), 25.8% (n=93) and 21.7% (n=78), respectively (Figure 1)”. My calculation was 21.6% instead of 21.7%.

c. I suggest specifying the % for each of the variables described in the last paragraph of page 15. It will maintain consistency with the other description of the results.

5. Discussion section: this section is in line with the results; the authors have discussed the data and brought the arguments. Few suggestions which follow:

a. “The main finding of the present study was a high prevalence (about 40%) of MetS in patients with chronic ChD that was greater than in the general population in other studies conducted in Brazil and worldwide (ref 10)”. Please add more references.

b. “Therefore, a possible explanation to the high prevalence of MetS observed in our

study is the high percentage of 61% of the participants aging >60 years.” It was previously mentioned in the beginning of this paragraph. I suggest removing it.

c. “The educational level was also an important variable related to MetS in our study, wherein those with >12 years of education had 64% lower odds to develop MetS in comparison to those with < 9 years (%)”. Please add the %.

d. “Surprisingly, carbohydrate consumption was a protective factor for the development of MetS in our study, although not reaching statistical significance in the multivariate model (OR 0.99; 95%CI 0.99 to 1.00; p=0.06)”. I did not find this p=0.06 in the table 2.

6. PLOS authors have the option to publish the peer review history of their article (what does this mean?). If published, this will include your full peer review and any attached files.

Reviewer #1: No

---

## [Author Response · Author response to Decision Letter 0]

23 Nov 2020

Journal Requirements:

Response: The manuscript was corrected to meet the Plos One`s style requirements.

2. Thank you for stating in your manuscript test that "All participants received information about the goals and procedures of the study and agreed to participate by signing an informed consent form." Please also add this information to your ethics statement in the online submission form.

 Response: This information was included in the ethics statement in the online form.

 Response: In the development of the present study, we used a physical form to store the information and the data were released on a digital platform Research Electronic Data Capture (REDCap). All the questionnaires included in the present study are validated tools widely used in the literature. The specifications on the variables included in the study are described in the methods section.

4. Please include a caption for figure 1.

 Response: We included the caption in figure 1, as suggested.

Additional comments to address:

Abstract: Line 9 starts with “Most participants were women (56.2%)”. The use of the term “Most” can be replaced with a different term as women participants are only slightly higher than men.

Response: The sentence has been replaced to "About half were female (56.2%)".

Data analysis: Since there were multiple comparisons, wondering if there was an attempt to correct for this by adjusting the P values and significance threshold?

Response: We appreciate the reviewer comment. Historically, correcting the p-values for multiple tests began with post-hoc testing following ANOVA. In theory, this rationale is also applied to all statistical procedures with multiple comparisons. However, considering that the number of tests performed in multiple regression analysis are usually high, it would be very difficult to declare a statistically significant result considering corrections for multiple tests. Therefore, the use of statistical tests to correct for multiple comparisons in multiple regression models is not common. A possible strategy to deal with multiple comparisons is to change the significance level to lower values (0.01, for example). In our study, all variables maintained in the multivariate model had p-values <0.01. Therefore, our results were consistent even if setting the probability of type 1 error to 1%. The multiple comparison issue was included as a possible limitation in the last paragraph of the discussion.

Results and Discussion: Majority of the study participants were from Mulatto ethnic group. It will be useful to discuss whether a high prevalence (40%) of MetS in patients with chronic ChD identified in this population is generalizable to Brazilian population broadly.

Response: The high prevalence (40%) of MetS found in the study can only be applied to the ChD population since they have specific characteristics. We included this information in the limitation paragraph of the discussion section.

In the last part of the discussion it will be useful to state how this study finding can be helpful in developing preventive strategies to minimize the risk of MetS in ChD patients.

Response: Thanks for the suggestion. We included some strategies that can be used to prevent and minimize the risk of MetS in patients with ChD. 

Reviewers' comments:

Introduction section: 

a. It will be better if the manuscript had line numbers and page references.

Response: The page and line numbers were included in the manuscript.

b. In the first paragraph of the introduction, reference 1 refers to an article from 2015 and reference 2 refers to an article from 2018 (Médecins Sans Frontières). I suggest using more updated information, such as the WHO website. 

Response: The reference was updated, as suggested by reviewer.

c. The MetS is mentioned for the first time in the second paragraph. I suggest reviewing this paragraph and better describe the syndrome, including a short history of the background and how there has been varying definitions until the WHO provided the currently recognized international definition.

Response: We have included a paragraph better describing the MetS, including the current WHO definition.

Methods section

1. If the patients are from different sites or a specific site in Brazil, it would beneficial to provide a map of the study site(s).

Response: The patients included in the study have different origins, most of whom are from rural areas in Brazil. However, at the time the study was conducted, all participants lived in the metropolitan region of the state of Rio de Janeiro. We included this specification in the study design, period and population section.

2. I would recommend separating the methods into:

a. Study design, period and population

b. Sample size. 

c. Study procedure or Inclusion criteria. 

d. Ethical considerations

e. Clinical follow up 

f. Clinical form of ChD

g. Evaluation of nutritional status

h. Socioeconomic data and lifestyle

i. Data management and statistical analysis 

Response: The methods were separated into new subsections.

Results section

a. Please specify in the text where this data can be found (table?) “The overall mean age was 60.7 years, with 56.2% women. There was a predominance of mulatto race (59.8%) and most participants had less than 9 years of schooling (67.3%)”. If it is listed in the table 1, please review the values, they are different from table 1.

Response: These data represents the major characteristics of the overall study sample and were included only in the manuscript text to avoid repeated data presentation. 

b. The percentage regarding “The prevalence of hypertension, dyslipidemia, obesity and diabetes were 67.3% (n=243), 53.5% (n=193), 25.8% (n=93) and 21.7% (n=78), respectively (Figure 1)”. My calculation was 21.6% instead of 21.7%.

Response: We apologize for this mistake. The correct value is 21.6%. This information was corrected in the manuscript and Figure 1.

c. I suggest specifying the % for each of the variables described in the last paragraph of page 15. It will maintain consistency with the other description of the results.

Response: In order to meet the reviewer`s request, we include the values for each variable described in the suggested paragraph.

Discussion section: 

a. “The main finding of the present study was a high prevalence (about 40%) of MetS in patients with chronic ChD that was greater than in the general population in other studies conducted in Brazil and worldwide (ref 10)”. Please add more references. 

Response: We have included three additional references in this paragraph.

b. “Therefore, a possible explanation to the high prevalence of MetS observed in our study is the high percentage of 61% of the participants aging >60 years.” It was previously mentioned in the beginning of this paragraph. I suggest removing it.

Response: We removed this information from the text.

c. “The educational level was also an important variable related to MetS in our study, wherein those with >12 years of education had 64% lower odds to develop MetS in comparison to those with < 9 years (%)”. Please add the %.

Response: The percentage included in this sentence refers to the protective effect of >12 years of education (64% lower odds of having MetS). We rephrased the sentence to improve clarity.

d. “Surprisingly, carbohydrate consumption was a protective factor for the development of MetS in our study, although not reaching statistical significance in the multivariate model (OR 0.99; 95%CI 0.99 to 1.00; p=0.06)”. I did not find this p=0.06 in the table 2.

Response: The p-value of 0.06 refers to the multivariate analysis; therefore, it is not shown in Table 2 that only includes values for the univariate analysis. The p-value of 0.06 (for carbohydrate) was also not seen in the Table 3 because it was not statistically significant.

---

## [Decision Letter · Decision Letter 1]

12 Mar 2021

Prevalence of metabolic syndrome and associated factors among patients with chronic Chagas disease

PONE-D-20-16190R1

Dear Dr. Mediano,

We’re pleased to inform you that your manuscript has been judged scientifically suitable for publication and will be formally accepted for publication once it meets all outstanding technical requirements.

Kind regards,

Dario Ummarino, Ph.D.

Senior Editor

PLOS ONE

Additional Editor Comments (optional):

Reviewers' comments:

Reviewer's Responses to Questions

**Comments to the Author**

1. If the authors have adequately addressed your comments raised in a previous round of review and you feel that this manuscript is now acceptable for publication, you may indicate that here to bypass the “Comments to the Author” section, enter your conflict of interest statement in the “Confidential to Editor” section, and submit your "Accept" recommendation.

Reviewer #1: All comments have been addressed

Reviewer #2: All comments have been addressed

2. Is the manuscript technically sound, and do the data support the conclusions?

Reviewer #1: Yes

Reviewer #2: (No Response)

3. Has the statistical analysis been performed appropriately and rigorously? 

Reviewer #1: Yes

Reviewer #2: (No Response)

4. Have the authors made all data underlying the findings in their manuscript fully available?

Reviewer #1: Yes

Reviewer #2: (No Response)

5. Is the manuscript presented in an intelligible fashion and written in standard English?

Reviewer #1: Yes

Reviewer #2: (No Response)

6. Review Comments to the Author

Reviewer #1: (No Response)

Reviewer #2: (No Response)

7. PLOS authors have the option to publish the peer review history of their article (what does this mean?). If published, this will include your full peer review and any attached files.

Reviewer #1: No

Reviewer #2: No

---

## [Editor Report · Acceptance letter]

24 Mar 2021

PONE-D-20-16190R1 

Prevalence of metabolic syndrome and associated factors among patients with chronic Chagas disease 

Dear Dr. Mediano:

I'm pleased to inform you that your manuscript has been deemed suitable for publication in PLOS ONE. Congratulations! Your manuscript is now with our production department. 

Kind regards, 

on behalf of

Dr. Dario Ummarino 

Staff Editor

PLOS ONE